# Machine Learning-Based Gene Expression Analysis to Identify Prognostic Biomarkers in Upper Tract Urothelial Carcinoma

**DOI:** 10.3390/cancers17162619

**Published:** 2025-08-11

**Authors:** Bernat Padullés, Ruben López-Aladid, Mercedes Ingelmo-Torres, Fiorella L. Roldán, Carmen Martínez, Judith Juez, Laura Izquierdo, Lourdes Mengual, Antonio Alcaraz

**Affiliations:** 1Urology Department and Laboratory, Hospital Clínic de Barcelona, 08036 Barcelona, Spain; padulles@clinic.cat (B.P.); lmengual@ub.edu (L.M.); aalcaraz@clinic.cat (A.A.); 2Genetics and Urological Tumors, Institut d’Investigacions Biomèdiques August Pi i Sunyer (IDIBAPS), 08036 Barcelona, Spain; 3Data Science Department at Premium Research, 19001 Guadalajara, Spain; ruben.aladid@gmail.com; 4Surgery and Surgical Specialities Department, Facultat de Medicina i Ciències de la Salut, Universitat de Barcelona (UB), 08036 Barcelona, Spain; 5Biomedical Sciences Department, Facultat de Medicina i Ciències de la Salut, Universitat de Barcelona (UB), 08036 Barcelona, Spain

**Keywords:** biomarker, gene expression, machine learning, random forest, prognosis, upper tract urothelial carcinoma

## Abstract

Upper tract urothelial carcinoma (UTUC) is a rare but aggressive cancer with limited tools to predict patient outcomes after surgery. This study applies machine learning tools for gene expression analysis to identify biomarkers that can help predict disease progression. By analyzing gene expression patterns in tissue samples from 17 UTUC patients who underwent surgery, we identified ten key prognostic genes. These findings highlight potential biomarkers for future research to improve patient care and outcomes.

## 1. Introduction

Upper tract urothelial carcinomas (UTUCs) are uncommon malignancies that arise in the lining of the renal pelvis and ureter, comprising only a small fraction (5–10%) of urothelial carcinomas [1]. Although rare, UTUCs are often diagnosed at more advanced stages compared to bladder cancer, partly due to their anatomical location and the lack of early, specific symptoms. In Western countries, UTUC incidence has been rising in recent decades, likely due to advancements in detection techniques and the aging population [2,3,4,5]. Given its aggressive nature, radical nephroureterectomy (RNU) stands as the established “gold standard” treatment for localized tumors [6,7].

To date, pathological stage and tumor grade are the only established prognostic factors associated with tumor progression and survival [8]; however, they are insufficient to predict the individual outcome of these tumors [9]. Based on pathological stage, pTa and pT1 tumors are often associated with a favorable prognosis, whereas pT4 pathology is usually associated with very limited survival. Meanwhile, pT2 and pT3 tumors constitute a significant portion of patients, with limited information available for predicting tumor progression. In the effort to improve patient outcomes, researchers and clinicians are increasingly interested in identifying new biomarkers with potential for early detection, treatment monitoring, and predicting disease progression. A deeper understanding of tumor biology could lead to the development of personalized treatment plans for patients.

The emergence of next-generation sequencing (NGS) technologies in recent years has significantly enhanced our understanding of the genetic alterations found in different solid tumors, including UTUC [10]. These techniques provide comprehensive DNA mutational and gene expression profiles, which are essential for stratifying patients based on their risk of progression [11].

Testing for these genetic alterations has become a routine part of clinical practice for various tumor types, complementing the traditional approach of determining cancer treatment based solely on clinicopathological and histological characteristics [12]. Recent studies have started to characterize the transcriptomic landscape of UTUC and identify potential prognostic markers [13,14,15,16,17,18]. However, available data remain scarce, fragmented, and often based on small cohorts or limited gene panels. There is a clear need for comprehensive approaches to uncover robust molecular predictors of disease progression in UTUC.

In this study, we propose the use of a supervised machine learning algorithm, specifically random forest, to analyze gene expression profiles derived from RNA sequencing of RNU tissue specimens from pT2 and pT3 UTUC patients, with the aim of identifying biomarkers with potential prognostic value. Our goal is to contribute to developing a gene expression-based prognostic tool that may aid in predicting tumor progression in UTUC patients. Integrating machine learning with transcriptomic profiling in a rare cancer such as UTUC represents an innovative approach that addresses a significant clinical gap and aims to uncover valuable molecular insights into this understudied malignancy. By doing so, we hope to fill a critical knowledge deficit in the current literature and contribute to advancing the understanding of UTUC biology.

## 2. Materials and Methods

### 2.1. Patients

Seventeen consecutive patients diagnosed with UTUC and treated at the Hospital Clínic of Barcelona were retrospectively included in this study. All patients underwent RNU with bladder cuff excision between 2004 and 2014. Inclusion criteria were (1) histologically confirmed UTUC, (2) pathological stage pT2 or pT3, and (3) availability of formalin-fixed paraffin-embedded (FFPE) tumor tissue suitable for RNA extraction. Exclusion criteria included the presence of other concurrent active malignancies at the time of diagnosis, incomplete clinical records, or insufficient tissue for transcriptomic analysis.

Demographic, pathological, and clinical data were collected from electronic medical records. All cases were reviewed by experienced genitourinary pathologists.

Clinical follow-up was carried out according to institutional guidelines, including abdominal and thoracic computed tomography (CT) scans every 3 months during the first year, every 6 months for the following two years, and annually thereafter. Tumor progression was defined as the development of local recurrence or distant metastases, confirmed radiologically or histologically. The date of progression was recorded as the date of radiologic or pathologic confirmation. Patients without progression were censored at the time of last follow-up. For comparative purposes, patients were stratified according to clinical progression status during follow-up (progressors vs. non-progressors).

This investigation was approved by the Clinical Research Ethics Committee of the Hospital Clinic of Barcelona (HCB/2018/0026).

### 2.2. Samples and RNA Extraction

FFPE tissue blocks were obtained from the IDIBAPS Tumor and Tissue Bank. The tumor area was macro-dissected from slides and RNA was isolated from four cuts of FFPE specimens (total thickness 80 µm) using the RecoverAll™ Total Nucleic Acid Isolation kit for FFPE (Ambion, Inc. Austin, TX, USA), following the manufacturers’ instructions. RNA was quantified by spectrophotometric analysis at 260 nm (NanoDrop Technologies, Wilmington, DE, USA), and RNA integrity was assessed using the Agilent 2100 Bioanalyzer System with the Eukaryote Total RNA Nano kit [19].

### 2.3. Library Preparation, Sequencing, Data Processing, and Analysis

RNA-seq libraries were prepared using Ion AmpliSeq™ Transcriptome Human Gene Expression Kit (Thermo Fisher Scientific, Waltham, MA, USA) following the manufacturer’s protocol; this enables the quantification of the expression levels of over 20,000 human RefSeqs. Briefly, cDNA was synthesized from total RNA using the SuperScript^®^ VILO™ cDNA Synthesis kit (Thermo Fisher Scientific) from 10 ng of RNA. Then, cDNA was amplified using Ion AmpliSeq™ technology. Finally, after partial digestion of the primer sequence with FUPA reagent, ligation of the barcoded adapters, and purification of the amplified cDNA using Agencourt^®^ AMPure^®^ XP Reagent, the library was quantified with the Ion Library TaqMan™ Quantitation Kit (Thermo Fisher Scientific). RNA sequencing was performed on an Illumina HiSeq 2500 sequencer, generating 100 bp single-end reads.

RNA-seq reads were aligned to the human reference genome (GRCh38) using STAR aligner v2.7.10a [20]. Gene counts were obtained with the FeatureCounts v2.0.3 software package [21].

### 2.4. Gene Expression and Functional Analysis

Data normalization and differential expression analysis between progressive and non-progressive patients were conducted using DESeq2 [22], with genes considered differentially expressed if they had an adjusted *p*-value (*p*-adj) < 0.05 and a log2 fold change. Volcano plots were generated to visualize differentially expressed genes using the Enhanced Volcano package in R [23]. To confirm the discriminative power of upregulated and downregulated genes, they were evaluated by means of linear logistic regression models using the glm function in R. To validate the models’ performance, receiver operating characteristic (ROC) curves were generated (for both upregulated and downregulated genes) using the pROC package in R [24]. Gene Ontology (GO) term enrichment analysis was performed using clusterProfiler [25], identifying significantly enriched GO terms among differentially expressed genes.

### 2.5. Machine Learning Explainability Model

To explore the genes most associated with cancer progression, a random forest classifier was applied to the differentially expressed genes identified between progressive and non-progressive patients [26]. This method was implemented using the randomForest package in R, with default parameters unless otherwise specified. The algorithm was selected due to its suitability for high-dimensional datasets and its relative robustness to overfitting, which is particularly important in studies with limited sample sizes (*n* = 17) [27].

Although the small cohort size limits the predictive generalizability of the model, the analysis was conducted with the aim of identifying candidate genes with potential biological relevance. Feature importance was evaluated using two standard metrics provided by the random forest algorithm: mean decrease in accuracy and Gini index. These measures quantify the contribution of each variable to the model’s classification performance.

To provide an approximate measure of model discrimination, overall accuracy, area under the receiver operating characteristic curve (AUC), sensitivity, precision, Matthews Correlation Coefficient (MCC) and F1-score were also calculated. These metrics were interpreted descriptively, given the exploratory nature of this study and the sample size limitations. Genes with the highest importance scores were retained for further biological interpretation as key contributors to UTUC progression [28].

## 3. Results

### 3.1. Clinicopathological Features of the Cohort

Overall, 17 UTUC patients (11 males, 6 females) with a median age of 74 years (range: 53–92) were included. The clinicopathological features of the enrolled patients are summarized in Table 1.

During a median follow-up of 36 months, seven (41%) patients progressed, with a mean time to progression of 14 months (range 4–37 months). Three of these seven patients had pT2 tumors, while the remaining four exhibited pT3 pathology. One patient had positive lymph nodes (LN+) at the time of RNU. All tumors were high-grade, which is representative of the general UTUC population undergoing radical nephroureterectomy.

All seven progressive patients died due to the UTUC. The mean cancer-specific survival (CSS) was 21 months (range 9–50 months).

### 3.2. Differential Gene Expression and GO Enrichment Analyses

Differential gene expression (DGE) analysis was conducted from RNA-seq data, assessing expression levels of a total of 11,673 genes. A total of genes were differentially expressed between progressive and non-progressive UTUC; 13 were upregulated, and 63 were downregulated in progressive compared with non-progressive UTUC. Differentially expressed genes between progressive and non-progressive UTUC are highlighted in the volcano plot (Figure 1) and Appendix A. 

The predictive capability of upregulated and downregulated genes was evaluated using a linear logistic regression. The logistic regression model for upregulated genes reached an AUC of 0.75, suggesting that this model can reasonably distinguish between significantly upregulated genes in progressive versus non-progressive UTUC patients (Figure 2).

The logistic regression model for downregulated genes had a high predictive accuracy (AUC; 0.95), confirming that the downregulated genes identified in this study exhibit significant predictive capability when evaluated using a logistic regression model.

GO enrichment analysis further provided insights into the biological processes significantly impacted within the dataset. The enriched GO terms included “Regulation of T-Cell Activation” (False Discovery Rate—FDR 0.02), “Positive Regulation of Lymphocyte Activation” (FDR 0.03), “Positive Regulation of Leukocyte Cell-Cell Adhesion” (FDR 0.03), “Integrin-Mediated Signaling Pathway” (FDR 0.03), and “Regulation of Endopeptidase Activity” (FDR 0.05). These terms indicate a pronounced involvement of immune-related processes in the DGE profile observed. Figure 3 illustrates the GO enrichment network, where differentially expressed genes are connected to the significantly enriched biological processes, highlighting shared functions and interactions.

### 3.3. Random Forest Classification for Disease Progression

The random forest algorithm identified a total of 10 differentially expressed genes as those contributing most significantly to UTUC progression. These genes and their relative influence on disease progression classification are depicted in Figure 4. Among them, *HLA-DQA1*, *HLA-DOB*, *ITM2A*, *THAP10*, *KRTAP5-3*, *CYP20A1*, *WNT5B*, and *MELK* were underexpressed, whereas *SPOCD1* and *TOMM6* were overexpressed (Appendix A). The importance of each gene in predicting disease progression was quantified using feature importance scores (ISs) derived from the random forest model. This score reflects each gene’s relative contribution to the model’s decision-making process. Among the selected genes, *HLA-DQA1* and *HLA-DOB* showed the highest importance scores (0.25 and 0.20, respectively), suggesting a strong influence on the classification outcome. Other genes such as *ITM2A*, *THAP10*, and *KRTAP5-3* also demonstrated moderate importance (ISs ranging from 0.08 to 0.15). Genes with lower importance scores, such as *SPOCD1* and *TOMM6* (IS = 0.04 and 0.03, respectively), were retained as they surpassed the threshold of 0.01 set for inclusion.

To assess the model’s overall discriminative performance, a ROC curve was generated using the full dataset, yielding an AUC of 0.88 (Figure 5). Additionally, five-fold cross-validation was performed to evaluate the model’s generalizability. The model achieved an average accuracy of 0.876 ± 0.152 and a precision of 0.800 ± 0.400. Sensitivity was 0.667 ± 0.422, and the F1-score reached 0.700 ± 0.400. The average AUC across cross-validation folds was 0.833 ± 0.211, indicating good discriminative ability. The MCC averaged 0.694 ± 0.403.

## 4. Discussion

Currently, there is a significant lack of information about which patients with UTUC are at risk of disease progression following RNU. Despite advancements in the management of these tumors, robust biomarkers that can reliably predict patient outcomes are still lacking; none have been successfully implemented in clinical practice. In this study, we applied machine-learning approaches for the analysis of UTUC gene expression profiles to identify potential robust biomarkers that could provide clinicians with more accurate prognostic information, ultimately improving decision-making and patient outcomes.

In our study, we identified ten genes with high predictive capability, suggesting they may serve as potential biomarkers of disease outcomes. These genes have previously been linked to cancer and are involved in metabolic processes associated with tumor progression. All of them play key roles in pathways related to cell cycle regulation, immune response, and metabolic reprogramming, which are crucial for cancer development and progression. Among them, *HLA-DQA1* and *HLA-DOB* are key regulators in the antigen presentation pathway, impacting immune surveillance. Dysregulation of these genes may impair antigen recognition, allowing tumors to evade immune detection. In accordance with our results, low *HLA-DQA1* expression was associated with poor prognosis in hepatocellular carcinoma, lung cancer, breast cancer, and soft tissue sarcoma; its reduction indicated the presence of an immunosuppressive microenvironment and invasive disease [29,30,31,32]. Similarly, low *HLA-DOB* expression in ovarian cancer was associated with a poor prognosis [33].

*ITM2A* and *SPOCD1* are involved in chromatin remodeling and cellular differentiation. In the context of cancer, alterations in *ITM2A* may contribute to tumor progression by affecting cellular responses to stress and altering the balance between cell survival and programmed cell death. In accordance with our results, the low expression of *ITM2A* in breast, ovarian, and cervical cancer tissues has been associated with poorer prognosis and advanced tumor stage [34,35,36,37]. A recent study underscored a significant correlation between low *ITM2A* expression in bladder cancer tissues and high tumor grade, pathological stage, and poor overall survival [38]. On the other hand, many studies demonstrated the upregulation of *SPOCD1* in different solid tumors [39,40,41]. In the same way, a high expression of *SPOCD1* in progressive UTUC patients was found in this study. Curiously, *SPOCD1* was downregulated in progressive bladder cancer patients [42], which is particularly interesting as it further supports the emerging molecular distinction between bladder cancer and UTUC [43].

*THAP10* and *MELK* are involved in controlling apoptosis and cell cycle progression. Downregulation of *THAP10* reduces apoptosis, favoring uncontrolled proliferation [44], while *MELK* overexpression is linked to aggressive tumor behavior and poor outcomes in solid cancers, including bladder cancer [45]. However, in our study on UTUC, *MELK* was underexpressed in progressive UTUC patients. This further supports the molecular divergence between UTUC and bladder cancer, as its downregulation in progressive UTUC contrasts with its known overexpression in aggressive bladder tumors.

*KRTAP5-3* and *WNT5B* contribute to epithelial cell stability and signaling. *KRTAP5-3* dysfunction can weaken epithelial barriers, facilitating metastasis [46]. *WNT5B* overexpression has been shown to enhance tumor invasiveness through the Wnt signaling pathway in some cancers [47]; however, the underexpression of this gene in our cohort could reflect a distinct molecular mechanism or a specific feature of UTUC, potentially influencing tumor behavior.

*CYP20A1* and *TOMM6* influence tumor metabolism and response to stress. While the exact role of *CYP20A1* in oncogenesis is unclear, changes in expression could potentially influence cancer progression through its involvement in metabolic pathways [48]. *TOMM6* plays a role in mitochondrial function, supporting metabolic adaptations in tumor cells [49].

Few studies in the literature focused on the study of UTUC gene expression. However, Fujii et al. [13] recently conducted an extensive molecular study of UTUC using integrated genomic analysis. They classified UTUC tissue specimens into five mutational subtypes with five specific expression subtypes (C1–C5), where C3–C5 were associated with a worse prognosis. These distinct molecular subtypes of UTUC align with the molecular alterations described in this study. The downregulation of *HLA-DQA1*, *HLA-DOB*, and *ITM2A*, which are associated with immune evasion and tumor progression, is consistent with the immune-excluded phenotype observed in the mesenchymal-like (C5) and aggressive basal/squamous-like (C4) subtypes. Similarly, the underexpression of *WNT5B* suggests a disruption in the Wnt signaling pathway, which may contribute to epithelial–mesenchymal transition, a hallmark of invasive UTUC subtypes. Moreover, the overexpression of *SPOCD1* and *TOMM6* in progressive UTUC patients may indicate metabolic and transcriptional reprogramming, characteristic of tumors within the basal/squamous (C4) and immune-infiltrated (C3) subtypes. These findings suggest that our differentially expressed genes may provide additional insights into the biological behavior of UTUC subtypes, reinforcing the relevance of molecular stratification for prognosis and potential therapeutic targeting.

Several other studies in the literature on UTUC patients analyzed gene expression and its association with tumor progression. However, these studies are limited by small sample sizes and the low number of genes analyzed. Mir et al. [14] demonstrated that *SLITRK6* expression is significantly higher in UTUC than in bladder carcinoma at both the mRNA and protein levels, positioning it as a promising therapeutic target for UTUC. Tomiyama et al. [15] found that high *TROP-2* expression was linked to favorable progression-free and cancer-specific survival in UTUC patients. Unfortunately, we found no significant differences between progressive and non-progressive disease for these genes in our cohort.

Our study stands out in UTUC research for utilizing high-throughput RNA sequencing for precise gene expression analysis, helping to identify putative new biomarkers. By applying machine-learning algorithms to UTUC gene expression data, we have been able to manage and interpret complex data, providing a powerful approach for identifying biomarkers for enhanced prognostic assessments and personalized treatments. In addition, GO enrichment analysis has enabled us to identify immune response pathways that may be suppressed in UTUC. This approach has allowed us to uncover correlations between gene expression and clinical outcomes in UTUC patients, offering more profound insights into tumor progression.

Despite the promising results, this study has some limitations, primarily the small sample size, which may affect the generalizability and robustness of the identified biomarkers. Given the high dimensionality of the data and limited sample size, we selected random forest as our primary machine learning algorithm, as it is well-suited for feature selection while minimizing the risk of overfitting; importantly, our aim was not to develop a predictive model, but rather to build an explanatory framework to identify genes potentially associated with cancer progression. Thus, although performance metrics such as sensitivity, precision, F1-score, and AUC are reported for completeness, they should be interpreted with caution, as predictive accuracy was not the main focus. Future studies with larger cohorts and external validation will be essential to confirm these findings and assess their potential clinical utility.

In this regard, external validation using the few available studies on gene expression in UTUC [13,14,15,16,17,18] was not feasible, as the underlying data were either not publicly available or otherwise inaccessible. To address this limitation, we are currently conducting a validation study using reverse transcription quantitative PCR in an independent cohort of UTUC patients from our institution. This study aims to confirm the prognostic value of the identified biomarkers in a separate patient population and through an alternative analytical approach. Given the low prevalence of UTUC, advancing biomarker research in this field will require coordinated, multi-institutional collaboration. Despite these challenges, we believe our findings offer valuable insights and contribute to laying a foundation for future studies in this underexplored area.

## 5. Conclusions

In this study, we demonstrate the feasibility and clinical potential of applying machine learning algorithms to transcriptomic data from UTUC patients to identify prognostic biomarkers. Ten genes emerged as robust candidates, each implicated in cancer-related pathways such as immune modulation, cell cycle regulation, and tumor progression. These findings not only contribute novel molecular insights into UTUC biology but also underscore the value of integrating computational models into biomarker discovery. While validation in larger and independent cohorts remains essential, this work lays the groundwork for the future development of personalized prognostic tools in UTUC management.

## Figures and Tables

**Figure 1 cancers-17-02619-f001:**
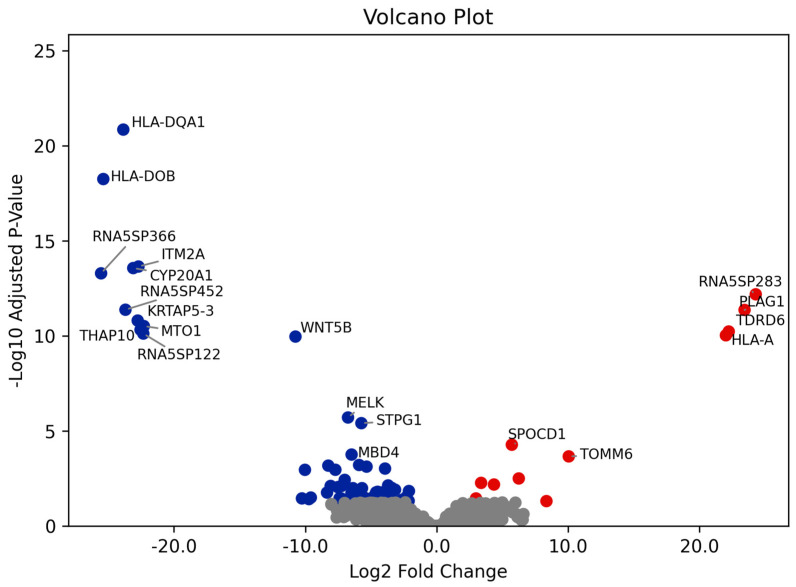
Volcano plot showing differentially expressed genes between progressive and non-progressive UTUC patients. The *x*-axis represents the Log2 fold change in gene expression levels, where positive values indicate upregulation in progressive UTUC patients (red points) and negative values indicate downregulation (blue points). The *y*-axis shows the −Log10 adjusted *p*-value, with higher values indicating stronger statistical significance.

**Figure 2 cancers-17-02619-f002:**
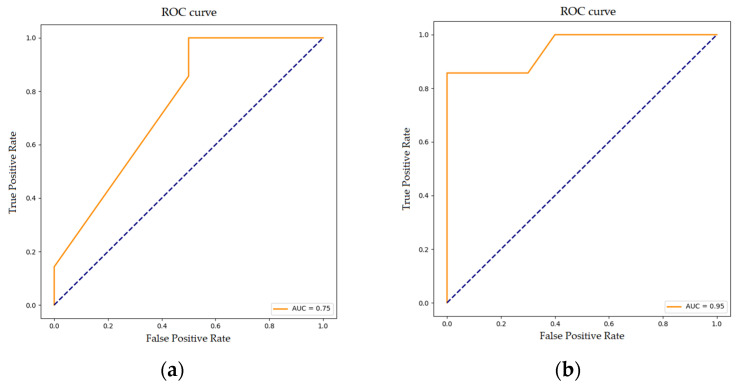
ROC curves illustrating the performance of the logistic regression model in predicting (**a**) upregulated and (**b**) downregulated genes.

**Figure 3 cancers-17-02619-f003:**
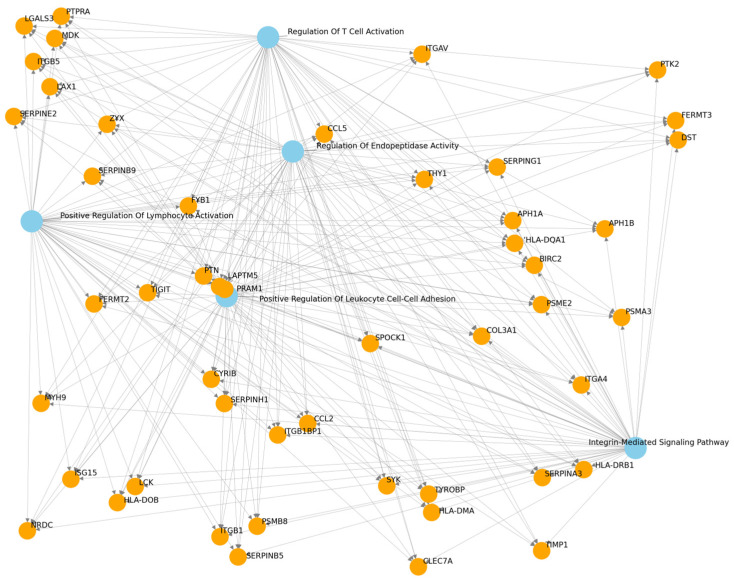
Network of pathways and gene terms. Orange nodes = differentially expressed genes; blue nodes = biological processes.

**Figure 4 cancers-17-02619-f004:**
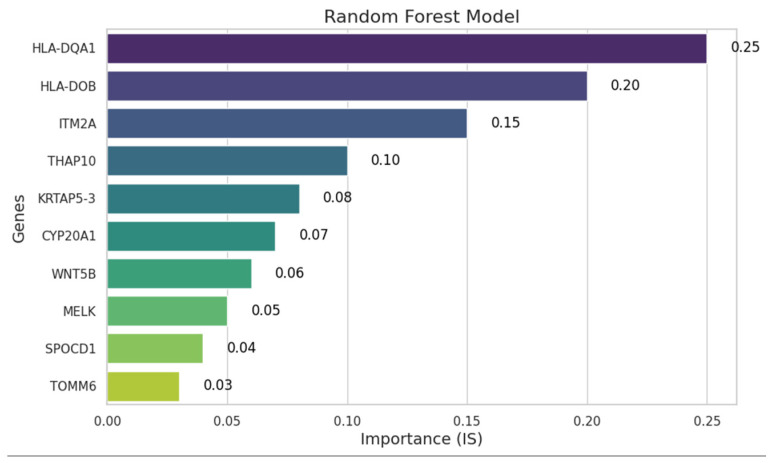
Feature importance plot for selected genes in a random forest model. The bars represent the importance score (IS) of each gene, indicating its relative contribution to the model. An IS of 0.01 was used as a cut-off.

**Figure 5 cancers-17-02619-f005:**
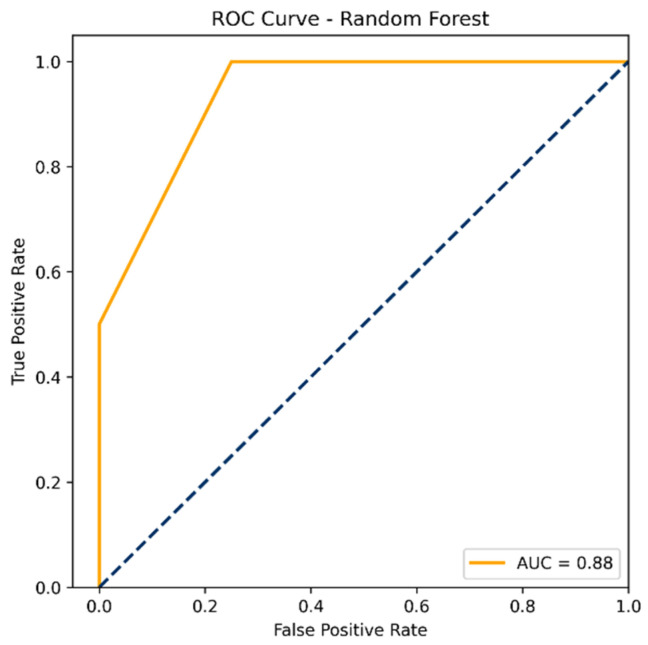
ROC curve for the random forest classifier using the top-ranked genes selected based on differential expression.

**Table 1 cancers-17-02619-t001:** Clinicopathological features of UTUC patients enrolled in this study.

	Progressive UTUC (*n* = 7)	Non-Progressive UTUC (*n* = 10)	Total UTUC (*n* = 17)
Gender, *n* (%)			
Male	4 (57)	7 (70)	11 (65)
Female	3 (43)	3 (30)	6 (35)
Tumor location, *n* (%)			
Pelvis	1 (14)	8 (80)	9 (53)
Ureter	4 (57)	1 (10)	5 (29)
Both	2 (29)	1 (10)	3 (18)
Pathological Stage, *n* (%)			
pT2	3 (43)	4 (40)	7 (41)
pT3	4 (57)	6 (60)	10 (59)
Histological Grade, *n* (%)			
Low	-	-	-
High	7 (100)	10 (100)	17 (100)
Metastasis, *n* (%)			
Local	-	-	-
Distant	5 (71)	-	5 (29)
Local + distant	2 (29)	-	2 (12)
Nodes, *n* (%)	1 (14)	-	1 (6)
Adjuvant Chemotherapy, *n* (%)	3 (43)	3 (30)	6 (35)

## Data Availability

Anonymized data can be made available from the corresponding author upon reasonable request and pending approval by the appropriate ethics committee.

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
