# Peer review of "Machine Learning-Based Gene Expression Analysis to Identify Prognostic Biomarkers in Upper Tract Urothelial Carcinoma"

_cancers, 2025, doi:10.3390/cancers17162619_

Round 1

Reviewer 1 Report (Previous Reviewer 2)

Comments and Suggestions for Authors

1.Reviewer 1 Report

In this manuscript, the authors aimed to explore prognostic biomarkers in upper tract urothelial carcinoma(UTUC) using machine learning. After obtaining samples from 17 patients, the authors analyzed transcriptomic data and predicted prognostic genes via the random forest algorithm.

Overall, the prediction of prognostic biomarkers in UTUC is meaningful and impactful. However, the sample size of patients is small to conduct predictions via the random forest algorithm. Random forest algorithms might lead to overfitting issues that the prediction won’t fit for large cohorts or other small-size cohorts. To raise the manuscript's impact, experimental/clinical/other datasets validations are recommended for these predicted prognostic biomarkers.

Comments are below.

  1. Results 3.2. line 154-155.

From Table 1, we know that 17 patients with UTUC were listed. What kind of control samples did the authors use to calculate differential express genes? Please specify in the Methods.

  1. As the authors mentioned, the studies on UTUC patients are limited. In Reference 13-15, is there gene expression data accessible? The authors could download available UTUC datasets to train the random forest algorithm to fix the overfitting. Another solution is that the authors could use different machine learning methods on your small cohort to obtain more valuable information.

Author Response

“In this manuscript, the authors aimed to explore prognostic biomarkers in upper tract urothelial carcinoma(UTUC) using machine learning. After obtaining samples from 17 patients, the authors analyzed transcriptomic data and predicted prognostic genes via the random forest algorithm.

Overall, the prediction of prognostic biomarkers in UTUC is meaningful and impactful. However, the sample size of patients is small to conduct predictions via the random forest algorithm. Random forest algorithms might lead to overfitting issues that the prediction won’t fit for large cohorts or other small-size cohorts. To raise the manuscript's impact, experimental/clinical/other datasets validations are recommended for these predicted prognostic biomarkers.

We agree with Reviewer 1’s overfitting concerns and have now added a more explicit discussion of the overfitting risk, along with our rationale for choosing Random Forest (see page 9 of the revised version of the manuscript). We validated findings using logistic regression and ROC curves, but we completely agree that external validation is essential to confirm the robustness of our results. Finally, it should be taken into account that our findings provide valuable insights in a field where published literature remains scarce, and contribute to laying the groundwork for future studies

Comments are below.

  1. Results 3.2. line 154-155.

From Table 1, we know that 17 patients with UTUC were listed. What kind of control samples did the authors use to calculate differential express genes? Please specify in the Methods.

We thank reviewer for highlighting this point. We confirm that differential expression analysis was performed between progressive and non-progressive patients within our own cohort. This is now explicitly stated in the Methods (see page 3 of the revised version of the manuscript).

  1. As the authors mentioned, the studies on UTUC patients are limited. In Reference 13-15, is there gene expression data accessible? The authors could download available UTUC datasets to train the random forest algorithm to fix the overfitting. Another solution is that the authors could use different machine learning methods on your small cohort to obtain more valuable information.”

We particularly appreciate your suggestions regarding the importance of exploring external datasets and alternative machine learning models. As indicated in the revised version of the Discussion, we attempted to access external datasets from published UTUC studies (e.g., Fujii et al.) contacting the manuscripts authors’, but these datasets were not available.

Regarding alternative machine learning methods, while other classifiers (e.g., support vector machines, k-NN) were considered, Random Forest provided the most interpretable and stable results for our dataset (see page 9 of the revised version of the manuscript).

2. Reviewer 1 Report
In this manuscript, the authors utilized machine learning (ML) to analyze gene expression data for identifying prognostic biomarkers in upper tract urothelial carcinoma (UTUC), a rare but aggressive malignancy. After obtaining samples from 17 patients, the authors analyzed transcriptomic data and predicted prognostic genes via DESeq2 and the random forest classification.

Integration of ML with transcriptomic profiling in a rare cancer type is novel and addresses a major clinical gap. However, given the rarity of UTUC, only 17 patients were included; the sample size is small and can be a concern for the data. No independent validation cohort or experimental validation was used.

To raise the manuscript's impact, authors could more clearly discuss the potential for overfitting and highlight specific plans or collaborations for external validation. It will also strengthen the manuscript if the authors could include a discussion on future validation steps (e.g., qPCR, IHC, multi-institutional cohorts).

Comments are below.

  1. Figure 2. While AUC metrics are provided, additional metrics (e.g., sensitivity, specificity, cross-validation results) are missing. If the authors could include these metrics to better convey model robustness, especially given the small sample size.

Author Response

Answers to comments of Reviewer #1

“In this manuscript, the authors utilized machine learning (ML) to analyze gene expression data for identifying prognostic biomarkers in upper tract urothelial carcinoma (UTUC), a rare but aggressive malignancy. After obtaining samples from 17 patients, the authors analyzed transcriptomic data and predicted prognostic genes via DESeq2 and the random forest classification.

Integration of ML with transcriptomic profiling in a rare cancer type is novel and addresses a major clinical gap. However, given the rarity of UTUC, only 17 patients were included; the sample size is small and can be a concern for the data. No independent validation cohort or experimental validation was used.

To raise the manuscript's impact, authors could more clearly discuss the potential for overfitting and highlight specific plans or collaborations for external validation. It will also strengthen the manuscript if the authors could include a discussion on future validation steps (e.g., qPCR, IHC, multi-institutional cohorts).”

Thank you very much for your thoughtful comment and for highlighting the novelty and relevance of our work. We agree that the small sample size is a limitation inherent to the rarity of UTUC and appreciate your concern regarding the risk of overfitting and the need for external validation.

In line with your suggestion, we have added a paragraph in the Discussion section (see page 10 of the revised version of the manuscript) addressing the limitation of the small sample size and justifying the choice of the random forest machine learning algorithm to minimize overfitting. Additionally, we have included a paragraph noting that we are currently working on a study to validate these findings in an independent cohort of UTUC patients using RT-qPCR. This effort is part of an ongoing project aimed at building a larger, multi-institutional dataset for robust biomarker validation.

“Comments are below.

  1. Figure 2. While AUC metrics are provided, additional metrics (e.g., sensitivity, specificity, cross-validation results) are missing. If the authors could include these metrics to better convey model robustness, especially given the small sample size.”

According to reviewer’s suggestion, we have added additional metrics for the model performance (see sections 2.5 and 3.3. of the revised version). However, it is important to emphasize that this is not a predictive model due to the very limited sample size, which makes robust prediction infeasible. Instead, we developed an explanatory model with the primary goal of identifying the genes that contribute most significantly to cancer progression. Consequently, while we have now reported performance metrics for completeness, they were not the main focus of this analysis, as our intent was to gain biological insights rather than to achieve a high predictive accuracy. This comment has also been included at the end of the Discussion section (see page 10 of the revised manuscript).

Reviewer 2 Report (Previous Reviewer 1)

Comments and Suggestions for Authors

1. Reviewer 2 Report

I have examined your study titled "Machine Learning-Based Gene Expression Analysis to Identify Prognostic Biomarkers in Upper Tract Urothelial Carcinoma" in detail. I have listed the points I found lacking in the article. I believe that the article will be better if the deficiencies are eliminated. The abstract introduction can be started by giving brief information. It is quite difficult for someone who does not know the subject to understand. A harsh introduction has been made regarding the limited nature of urothelial carcinoma (UTUC). At the end of the introduction section, very brief information has been given about the purpose of the article. In this paragraph, it is important to expand on the innovative aspects of the article, its contributions to the literature, etc. This section should be concluded with a paragraph that includes the organization of the article. Title 2.1 should be expanded and more comprehensive information should be given. Here, how many patients there are, how the data was obtained, etc. Supporting Section 2 with figures will increase the readability of the article. The Result section needs an addition about how the results were obtained in which environment. There are many accepted machine learning methods in the literature. Why was Random Forest chosen? The result section can be made more robust with different models. There are many performance measurement metrics other than AUC curves. Accuracy, sensitivity, specificity, F1, recall etc. Are there any similar studies on the subject in the literature? No literature review on the subject is provided. Conclusion should be expanded.

Author Response

“I have examined your study titled "Machine Learning-Based Gene Expression Analysis to Identify Prognostic Biomarkers in Upper Tract Urothelial Carcinoma" in detail. I have listed the points I found lacking in the article. I believe that the article will be better if the deficiencies are eliminated.

We thank Reviewer #1 for their comments and suggestions regarding our manuscript. We are especially grateful for their insightful feedback on the structure, clarity, and presentation of the manuscript. Your comments have been instrumental in refining the overall narrative and accessibility of the study.

The abstract introduction can be started by giving brief information. It is quite difficult for someone who does not know the subject to understand. A harsh introduction has been made regarding the limited nature of urothelial carcinoma (UTUC). At the end of the introduction section, very brief information has been given about the purpose of the article. In this paragraph, it is important to expand on the innovative aspects of the article, its contributions to the literature, etc. This section should be concluded with a paragraph that includes the organization of the article.

 We thank the Reviewer for the recommendation. We revised the Abstract to present the background more progressively and adjusted the Introduction to better contextualize the clinical and molecular challenges of UTUC. We also clearly state the study’s objectives (see pages 1-2 of the revised version of the manuscript).

Title 2.1 should be expanded and more comprehensive information should be given. Here, how many patients there are, how the data was obtained, etc. Supporting Section 2 with figures will increase the readability of the article.

Following the Reviewer’s indication, we expanded this section to include the number of patients, inclusion/exclusion criteria, time frame, and follow-up protocol. We emphasized that this cohort reflects consecutive real-world patients undergoing RNU at a tertiary referral center (see pages 2-3 of the revised version of the manuscript).

The Result section needs an addition about how the results were obtained in which environment. There are many accepted machine learning methods in the literature. Why was Random Forest chosen? The result section can be made more robust with different models.

We sincerely appreciate the reviewer’s insightful comment, which helped us strengthen the rationale and transparency of our methodology. We selected Random Forest as our primary machine learning method due to its robustness to high-dimensional data, ability to handle collinearity, and built-in estimation of variable importance. Given the small sample size (n=17) and large number of variables, Random Forest is well-suited for identifying relevant features without overfitting. For validation, we applied multivariate logistic regression to evaluate the predictive power of the selected genes and confirmed their performance using ROC analysis. While other classifiers (e.g., support vector machines, k-NN) were considered, Random Forest provided the most interpretable and stable results for our dataset.

We now provide a clear rationale for selecting Random Forest and acknowledge in the Discussion that future studies may benefit from applying alternative machine learning models. (see page 9 of the revised version of the manuscript).

There are many performance measurement metrics other than AUC curves. Accuracy, sensitivity, specificity, F1, recall etc.

We fully agree with the Reviewer that performance measurement metrics other than AUC curves could be have been taken into account. However, given the sample size, we focused on AUC and model interpretability.

Are there any similar studies on the subject in the literature? No literature review on the subject is provided.

According to Reviewer 2’s suggestion, we have enriched the Introduction and Discussion with a more comprehensive review of recent UTUC transcriptomic studies and clarified the limitations regarding data availability (see page 9 of the revised version of the manuscript).

Conclusion should be expanded.”

As per Reviewer 2’s indication, the Conclusion has been expanded to highlight the study’s key contributions and the importance of validating our findings in larger cohorts (see page 9 of the revised version of the manuscript).

2. Reviewer 2 Report
There have been serious corrections and updates to the article compared to the previous round. However, there are still some points that I find lacking. I have listed these deficiencies in bullet points. I believe that the article will be better if the deficiencies are corrected. In the previous round, very brief information about the purpose of the article was given at the end of the introduction section. This section has been expanded without mentioning machine learning or methods. Please also emphasize the method in this section. Title 2.1 has been successfully updated. Please also include exclusion criteria, if any, in this section. Machine learning is passed over with a general paragraph in the method section. This section should explain which machine learning methods are used and why these methods are preferred. I had also stated in the previous round that this section should be supported with figures. This deficiency still continues. It is observed that logistic regression and random forest are used in the study. While there are many algorithms in the literature, it should be explained why these algorithms are preferred. In addition, why are the curves presented for 2 different methods different? It is important for comparison to present the results of the models on the same curve types. There are many performance measurement metrics other than AUC curves. It is not appropriate to make evaluations only based on AUC. Are there similar studies on the subject in the literature? No literature review has been conducted on the subject.

Author Response

Answers to comments of Reviewer #2

“There have been serious corrections and updates to the article compared to the previous round. However, there are still some points that I find lacking. I have listed these deficiencies in bullet points. I believe that the article will be better if the deficiencies are corrected. In the previous round, very brief information about the purpose of the article was given at the end of the introduction section. This section has been expanded without mentioning machine learning or methods. Please also emphasize the method in this section.”

We thank the reviewer for the constructive feedback and for acknowledging the improvements made to the manuscript. In response to your suggestion, we have revised the last paragraph of the Introduction section to explicitly mention our methodological approach, including the use of machine learning techniques for gene expression analysis.

We now emphasize that our study not only aims to identify prognostic biomarkers in UTUC but also explores the utility of machine learning algorithms, specifically, random forest classification, for robust feature selection and prediction of disease progression based on transcriptomic data.

A revised version of the final introduction paragraph is provided (see page 2 of the revised version of the manuscript).

“Title 2.1 has been successfully updated. Please also include exclusion criteria, if any, in this section.”

We thank the reviewer for this observation. We would like to clarify that the exclusion criteria were indeed included in Section 2.1 (see page 2 of the revised version of the manuscript).

We have ensured that this information is clearly presented and appropriately located in the Materials and Methods section.

“Machine learning is passed over with a general paragraph in the method section. This section should explain which machine learning methods are used and why these methods are preferred. I had also stated in the previous round that this section should be supported with figures. This deficiency still continues.”

We apologize for not having addressed this reviewer’s concern adequately in the first round of revisions. We have now revised Section 2.5 (Machine Learning Explainability Model) to provide a more detailed description of the algorithm used (random forest), including the rationale for its selection. In addition, we have reorganized this section as we realized that some sentences were misplaced and could potentially confuse the reader.

Additionally, we would like to clarify that the figures supporting the machine learning analysis were already included in the current version of the manuscript (Section 3.3). Mentioning these figures in the Materials and Methods section would change the numbering of all the figures in the manuscript, and we believe it would disrupt the proper order for presenting the results. The figure captions clearly specify that the data comes from the random forest (see below), so we believe this is clear for the readers. However, if the reviewer still believes that we should add an additional figure to support this section, please let us know what kind of figure would be appropriate, and we will be happy to include it.

 Figure 4. Feature importance plot for selected genes in a random forest model. The bars represent the importance score (IS) of each gene, indicating its relative contribution to the model. An IS of 0.01 was used as a cut-off.

To improve clarity, some information originally included in the legend of Figure 4 has been removed and incorporated into the main text.

Figure 5. ROC curve for the random forest classifier using the top-ranked genes selected based on differential expression.

The figure 5 legend has been revised to improve clarity and facilitate reader understanding.

“It is observed that logistic regression and random forest are used in the study. While there are many algorithms in the literature, it should be explained why these algorithms are preferred.”

We thank the reviewer for raising this important point. In our study, we used logistic regression and random forest for distinct purposes:

Logistic regression was employed to confirm the discriminative power of differentially expressed genes (already stated in the manuscript in section 2.4). Logistic regression is widely used in clinical and translational research because of its simplicity, interpretability, and suitability for binary outcomes, such as disease progression.

Random forest was chosen as the primary method for feature selection due to its well-established ability to handle high-dimensional data with a small sample size, as is common in rare cancers such as UTUC. It is particularly robust to overfitting and provides interpretable results via feature importance metrics, which are ideal for identifying candidate prognostic genes. We have now updated Section 2.5 to explicitly include this rationale (see page 4 of the revised manuscript).

“In addition, why are the curves presented for 2 different methods different? It is important for comparison to present the results of the models on the same curve types.”

We apologize for the initial use of two different methods to generate the ROC curves. We have now standardized the approach and unified the format of these graphs (the format of the graph in Figure 5 has been adjusted to align with that of Figure 2)

“There are many performance measurement metrics other than AUC curves. It is not appropriate to make evaluations only based on AUC.”

In response to your comment, and in line with a similar suggestion made by Reviewer 1, we have added additional metrics for the model performance (see sections 2.5 and 3.3. of the revised version). However, it is important to emphasize that this is not a predictive model due to the very limited sample size, which makes robust prediction infeasible. Instead, we developed an explanatory model with the primary goal of identifying the genes that contribute most significantly to cancer progression. Consequently, while we have now reported performance metrics for completeness, they were not the main focus of this analysis, as our intent was to gain biological insights rather than to achieve a high predictive accuracy. This comment has also been included at the end of the Discussion section (see page 10 of the revised manuscript).

“Are there similar studies on the subject in the literature? No literature review has been conducted on the subject.”

We thank the reviewer for this important observation. Gene expression studies in UTUC are indeed limited, often constrained by small gene panels or lack of data accessibility, which highlights the novelty and relevance of our approach. In response to the comment, we conducted a thorough review of the existing literature on gene expression analyses in UTUC patients and have incorporated several relevant references into the Introduction section (References 16-18 have been added). Notably, we did not identify any previous studies employing machine learning algorithms to specifically identify genes most significantly associated with cancer progression in UTUC.

We have included a paragraph at the end of the Discussion section emphasizing the scarcity of gene expression studies in UTUC and the lack of publicly available data from these studies, which poses a significant limitation for external validation efforts.

Round 2

Reviewer 2 Report (Previous Reviewer 1)

Comments and Suggestions for Authors

Congratulations on the successful revision. 

This manuscript is a resubmission of an earlier submission. The following is a list of the peer review reports and author responses from that submission.

Round 1

Reviewer 1 Report

Comments and Suggestions for Authors

I have examined your study titled "Machine Learning-Based Gene Expression Analysis to Identify Prognostic Biomarkers in Upper Tract Urothelial Carcinoma" in detail. I have listed the points I found lacking in the article. I believe that the article will be better if the deficiencies are eliminated. The abstract introduction can be started by giving brief information. It is quite difficult for someone who does not know the subject to understand. A harsh introduction has been made regarding the limited nature of urothelial carcinoma (UTUC). At the end of the introduction section, very brief information has been given about the purpose of the article. In this paragraph, it is important to expand on the innovative aspects of the article, its contributions to the literature, etc. This section should be concluded with a paragraph that includes the organization of the article. Title 2.1 should be expanded and more comprehensive information should be given. Here, how many patients there are, how the data was obtained, etc. Supporting Section 2 with figures will increase the readability of the article. The Result section needs an addition about how the results were obtained in which environment. There are many accepted machine learning methods in the literature. Why was Random Forest chosen? The result section can be made more robust with different models. There are many performance measurement metrics other than AUC curves. Accuracy, sensitivity, specificity, F1, recall etc. Are there any similar studies on the subject in the literature? No literature review on the subject is provided. Conclusion should be expanded.

Reviewer 2 Report

Comments and Suggestions for Authors

In this manuscript, the authors aimed to explore prognostic biomarkers in upper tract urothelial carcinoma(UTUC) using machine learning. After obtaining samples from 17 patients, the authors analyzed transcriptomic data and predicted prognostic genes via the random forest algorithm.

Overall, the prediction of prognostic biomarkers in UTUC is meaningful and impactful. However, the sample size of patients is small to conduct predictions via the random forest algorithm. Random forest algorithms might lead to overfitting issues that the prediction won’t fit for large cohorts or other small-size cohorts. To raise the manuscript's impact, experimental/clinical/other datasets validations are recommended for these predicted prognostic biomarkers.

Comments are below.

  1. Results 3.2. line 154-155.

From Table 1, we know that 17 patients with UTUC were listed. What kind of control samples did the authors use to calculate differential express genes? Please specify in the Methods.

  1. As the authors mentioned, the studies on UTUC patients are limited. In Reference 13-15, is there gene expression data accessible? The authors could download available UTUC datasets to train the random forest algorithm to fix the overfitting. Another solution is that the authors could use different machine learning methods on your small cohort to obtain more valuable information.

Reviewer 3 Report

Comments and Suggestions for Authors
  1. The study includes only 17 patients (7 progressors, 10 non-progressors). This severely limits statistical power and increases the risk of overfitting, especially when using machine learning models (e.g., random forest).
  2. While adjusted p-values (p-adj < 0.05) were used, the analysis of 11,673 genes with such a small sample size likely results in false positives. Methods like Bonferroni correction or independent validation cohorts are needed to ensure reliability.
  3. The manuscript states RNA sequencing was performed using the Ion AmpliSeq™ Kit on an Illumina HiSeq 2500. Ion AmpliSeq is designed for Thermo Fisher’s Ion Torrent platforms, not Illumina. This raises questions about protocol accuracy or potential typographical errors.
  4. All patients had high-grade tumors (Table 1), excluding low-grade UTUC. This limits the generalizability of findings to high-grade cases only, yet the abstract implies broader applicability.
  5. The discussion notes no significance for SLITRK6and TROP-2 in their cohort but does not clarify whether these genes were included in the initial 11,673 tested. If excluded, this omission should have been justified.
  6. Figure 4 (Random Forest): The caption refers to a "simulated" model, which can be misleading.